# The Intestinal Microbiome and the Metabolic Syndrome—How Its Manipulation May Affect Metabolic-Associated Fatty Liver Disease (MAFLD)

**Stephen D. H. Malnick [1,2,\*] and Sheral Ohayon Michael [1,2]**

1   Department of Internal Medicine C, Kaplan Medical Center, Rehovot 76100, Israel; sheral90@gmail.com
2   Faculty of Medicine, The Hebrew University of Jerusalem, Jerusalem 91121, Israel
\*   Correspondence: stephen@malnick.net

**Abstract:** Metabolic-associated fatty liver disease (MAFLD) is now the predominant liver disease worldwide consequent to the epidemic of obesity. The intestinal microbiome (IM), consisting of the bacteria, fungi, archaea, and viruses residing in the gastrointestinal tract, plays an important role in human metabolism and preserving the epithelial barrier function. Disturbances in the IM have been shown to influence the development and progression of MAFLD and play a role in the development of metabolic syndrome (MS). The main treatment for MAFLD involves lifestyle changes, which also influence the IM. Manipulation of the IM by fecal microbial transplantation (FMT) has been approved for the treatment of recurrent *Closteroides difficile* infection. This may be administered by endoscopic administration from the lower or upper GI tract. Other methods of administration include nasogastric tube, enema, and oral capsules of stool from healthy donors. In this narrative review, we elaborate on the role of the IM in developing MS and MAFLD and on the current experience with IM modulation by FMT on MAFLD.

**Keywords:** fatty liver; intestinal microbiome; fecal microbial transplantation

## 1. Introduction

### 1.1. Metabolic-Associated Liver Disease and Metabolic Syndrome

Metabolic-associated fatty liver disease (MAFLD) is the hepatic evidence of metabolic syndrome. Metabolic syndrome (MS) is identified by central obesity, hypertension, hyperlipidemia, and insulin resistance. The National Cholesterol Education Program (NECP) defines MS as any three or more of the following: (1) fasting blood glucose greater than 100 mg/dL or drug treatment for elevated blood glucose; (2) HDL cholesterol < 40 mg/dL in men and 50 mg/dL in women or drug treatment for low HDL cholesterol; (3) serum triglycerides > 150 mg/dL or drug treatment of elevated triglycerides; (4) waist circumference > 102 cm in men or 88 cm in women; and (5) blood pressure > 130/85 mm Hg or drug treatment for hypertension [1]. There are differences in the definitions between Caucasian [2] and Asian populations [3], reflecting differences in body anthropomorphy. Due to the epidemic of obesity, there has been a precipitous increase in the prevalence of MS and MAFLD [4]. The percentage of the population with MAFLD in the USA is now higher than in Western Europe and Australasia. The highest prevalence is in Latin America, reaching up to 59% of the population [5]. There is no pharmacological treatment for MAFLD that has been shown to be superior to the established lifestyle changes, including weight loss, 150 min of walking weekly, and eating a Mediterranean diet. It is recommended to avoid alcohol and sweet sugary beverages and consume 3 cups of coffee a day and daily aspirin for the prevention of fibrosis [6]. Bariatric surgery may be effective, although there are data showing a long-term average of 28% weight gain after surgery [7].

The pathophysiology of MS is not completely understood. Central adiposity and insulin resistance are critically important [8]. The intestinal microbiome has an important role in the development of MS.

### 1.2. Intestinal Microbiome

The gut microbiome is made up of bacteria, fungi, archaea, and viruses [3]. A healthy microbiome has an important role in preserving epithelial barrier function [9–12]. The intestinal bacteria metabolize unabsorbed carbohydrates to short-chain fatty acids (SCFA), which are an important energy source [13]. Disturbances in the intestinal microbiome can result in a decrease in the small intestinal barrier function [14]. Alterations in the gut microbiota can influence the physiological function of the gut [15,16]. A state of dysbiosis exists when there is an alteration in the small intestinal bacteria. Dysbiosis impacts the host's immune response and physiological function via interactions between the microbiota and the food metabolites.

Bacterial overgrowth may develop because of intestinal tract secretions, including gastric and bile acid, mucin production, and gut antibacterial peptides. In addition, retrograde movement of bowel contents to the upper gut is inhibited by the ileocecal valve.

Microbial signaling refers to the translocation of bacterial metabolites or structural components of intestinal epithelial cells, which enables communication with other organ systems. The altered microbial signaling in dysbiosis results in changes in both GI motility and metabolism. Pathogen-associated molecular patterns (PAMPS), including LPS, peptidoglycans, and flagellin are detected by pattern recognition receptors. The aryl hydrocarbon receptor (AHR) is a transcription factor that affects responses to external stimuli. In the presence of dysbiosis, there are microbiota profiles that do not generate the AHR ligand, and this results in metabolic disorders [17,18]. Thus, dysbiosis can result in metabolic changes. Metabolic inflammation occurs because of changes in the gut barrier [16,19,20]. Furthermore, the gut microbiota produces signaling molecules, which influence the production of energy from indigestible carbohydrates [21,22] (Figure 1).

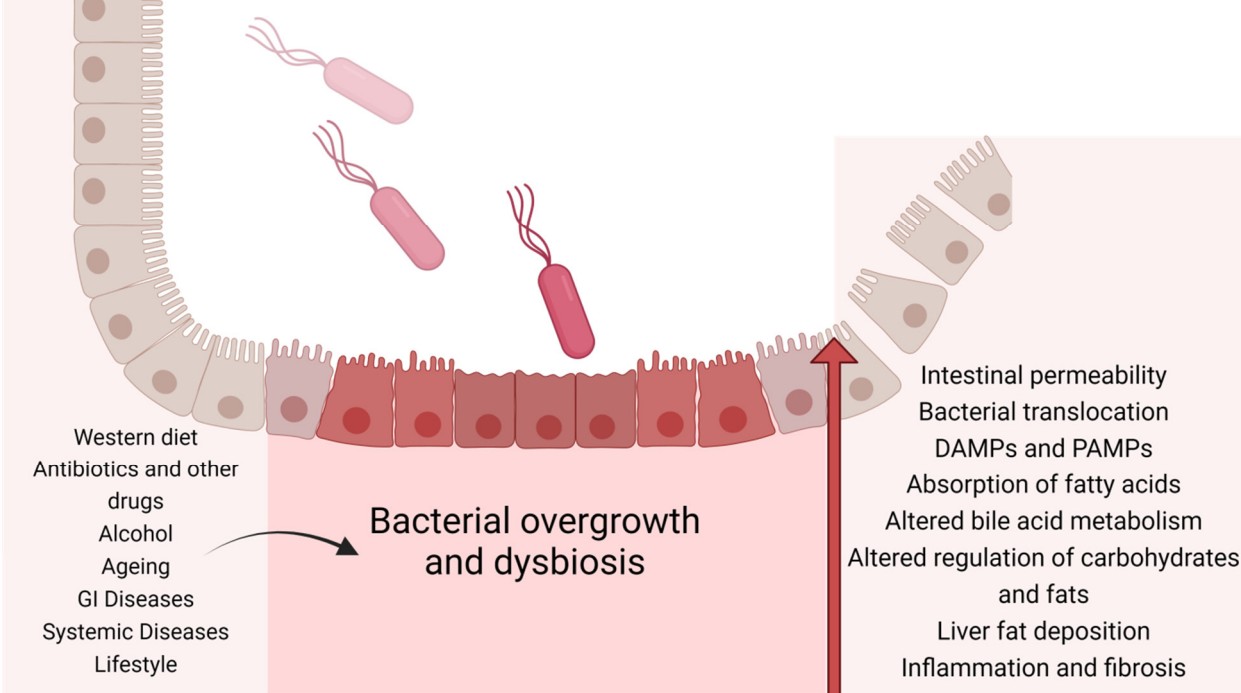

**Figure 1.** The mechanisms by which intestinal dysbiosis may influence the development of liver disease.

## 2. MAFLD

### 2.1. Dysbiosis

The microbiota of the gut regulates metabolic processes [23] and dysbiosis, which is associated with MS, affecting metabolic homeostasis, inflammation, and immunity [24]. Several mechanisms have been proposed for this, including lipid uptake by epithelial cells, hepatic gluconeogenesis, and insulin signaling. A full discussion of the evidence is beyond the scope of this review and has been recently summarized [25].

Antibiotic administration, which of course alters the microbiome, is linked to metabolic disease [26]. Early experiments in the 1950s in farm animals showed that growth was promoted by sub-therapeutic doses of antibiotics [27]. More recently, it has been shown that sub-therapeutic antibiotic treatment given to laboratory mice increased the body weight, worsened metabolic outcomes, and altered the gut microbiota [28]. Furthermore, the obese phenotype was transferred to germ-free mice after the administration of stools from mice treated with antibiotics. Additionally, antibiotic consumption in obese patients decreased peripheral insulin resistance [29]. These findings implicate the intestinal microbiome in the development of MS.

MS is defined by obesity, an increased abdominal circumference, hypertension, hyperlipidemia, and insulin resistance [30]. MAFLD severity increases as follows: macrovesicular steatosis to steatohepatitis and fibrosis, resulting in cirrhosis, including hepatocellular carcinoma. Due to the success of vaccination and treatment for chronic HBV hepatitis, the availability of direct-acting antivirals for chronic HCV hepatitis, and the epidemic of obesity, MAFLD is now one of the major causes of cirrhosis worldwide [31].

### 2.2. A Dysbiotic Gut Microbiome Is Associated with MS

Compositions of the gut flora of both human and animal models have been examined in order to further understand the effect of the microbiome on the development of MS. A Western diet that is dense with a high content of saturated fat and carbohydrates and low in fiber is recognized as a major cause of MS. There are many reports of an increased amount of *Bacteroides* and a lower level of *Pravotella* [32]. This was also present in immigrants to the USA from Southeast Asia [33].

The HELIUS study of the microbial profile in a multi-ethnic population detected a higher proportion of Enterobacteriae and a lower proportion of Peptostreptococcacae in patients with MS [34]. Other studies have reported different changes in the components of the microbiome [35,36].

The complexity of the influence of the microbiome and its interaction with MS was demonstrated in a study by Atzeni et al., who found a negative association between insulin resistance and *Desulfovibri* and *Odoribacter* and a positive association with *Feacalibatrum* and *Butyricicoccus* [37].

Another group reported a strong link between *Escherichia coli* and visceral obesity, which is an important component of metabolic syndrome [24].

The gut microbiota has an important role in the pathogenesis of MAFLD. The polysaccharides metabolize into monosaccharides. The dietary fibers break down into short-chain fatty acids (SCFA), and this provides an important source of energy to the intestinal mucosa. In addition, SCFAs have a role in promoting colonic motility, protecting the intestinal mucosal barrier, immune regulation, as well as anti-inflammatory actions. In both germ-free mice and mice treated with antibiotics, which depletes the microbiome, a lower proliferation of intestinal epithelial cells occurred. In addition, a decreased availability of SCFAs alters the energy in cells from the colon towards glucose utilization.

In an additional study, SCFAs activated glucagon protein-coupled receptors (GPRs) 41 and 43. GPR41 increases glucagon-like peptide 1 and peptide YY secretion from enteroendocrine cells, resulting in a decrease in intestinal mobility and enhanced absorption. GPR43 inhibits adipocyte differentiation, increases hepatic lipogenesis, and promotes the development of MAFLD.

Bile acids interact with the microbiome and have a role in the pathogenesis of MAFLD. Primary bile acids are agonists of FXR. They maintain intestinal microbiota homeostasis by inhibiting the overgrowth of pathogenic bacteria. This results in the activation of genes that protect the ileal mucosa. Furthermore, FXR produces the downregulation of both LXR and SREBP-1c, resulting in a reduction of hepatic fatty acid and triglyceride and glycogen synthesis, and a reduction in steatogenesis and gluconeogenesis.

Secondary bile acids also activate TGR5 (Takeda G protein receptor 5). The activation of TGR5 by secondary bile acids results in the conversion of thyroid hormone to tri-iodothyronine and increases basal metabolism. By promoting energy metabolism in the brown adipose and muscle tissues of mice fed a high-fat diet, there is an elevation of basal metabolism.

In addition, gut-derived metabolites also have a role to play. A study from Finland has reported correlations between circulating metabolites and gut microbiota composition [38]. Tryptophan metabolites derived from bacteria play an important role in the development of MS. They influence both the differentiation and function of anti-inflammatory cells, including T-regulating (Treg) cells [39]. Blood tryptophan metabolite levels were lower in patients with type 2 DM compared to controls [25].

The microbiome changes in patients with MS may also be linked with increased inflammation involving a decrease in short-chain fatty acid (SCFA) production [35].

Trimethylamine N-oxide (TMAO) in plasma has been associated with coronary artery disease and stroke. There has been a positive association reported between plasma levels and the abundance of Peptococcacae and *Preveotella* and a negative association with *F. prausnitzii* [38].

### 2.3. Intestinal Microbiome and Development of MAFLD

The intestinal microbiome plays an important role in the development of MAFLD. This has been shown in laboratory studies on mice. Human feces from an obese twin when transferred to germ-free mice resulted in a metabolic profile characteristic of obesity. However, when the mice were co-housed with mice that had been exposed to feces from a lean twin, the effect was weakened [40]. In addition, germ-free mice gained less weight while consuming a high-sugar and high-fat diet as compared to normal mice. This effect was maintained even when they consumed a larger amount of food [41].

Further evidence supporting the important role of the intestinal microbiome in the development of obesity and insulin resistance includes alteration of the insulin resistance index by FMT [42], which also alters fat accumulation in macrophages and glucose metabolism [42]. Antibiotic administration to mice with diet-induced obesity resulted in a decrease in fasting glucose levels and insulin resistance that was independent of food consumption and degree of adiposity [43]. Furthermore, antibiotic administration decreased both hepatic lipogenesis and steatosis [43].

In human studies, transfer of the intestinal microbiome from lean male donors to males with metabolic syndrome has been shown to decrease insulin resistance [44]. Furthermore, the intestinal microbiome of obese individuals has a different microbial signature and diversity compared to lean people [45]. More recently, it has been shown that there are specific microbiome signatures in both obesity and type 2 diabetes mellitus. This has been reviewed elsewhere [46]. Interestingly in NAFLD patients with advanced fibrosis, there appears to be a specific microbiome signature [47].

The relationship between the IM and the liver is complex. The liver receives 70% of its blood supply from the hepatic portal vein directly from the gastrointestinal tract. Thus, the liver is the first organ to be exposed to the microbiota from the gastrointestinal tract. This is a bidirectional relationship. Animal experiments involving bile duct ligation have shown changes in the intestinal microbiota and the permeability of the intestine [48]. Furthermore, in patients with cirrhosis, there has been an increase in intestinal, bacteria, translocation, and in circulating bacterial DNA fragments [49]. These observations demonstrate that

changes in the microbial composition of the gastrointestinal tract may not be a consequence of the changes in the liver, but rather a result of liver disease itself.

As described above, the gut microbiome is linked to obesity, insulin resistance, and liver steatosis. There is increasing effort to investigate interventions, targeting the gut microbiota in humans to modify or treat MAFLD. These include the administration of antibiotics, probiotics, prebiotics, synbiotics, postbiotics, or fecal microbiota transplantation (FMT).

## 3. Antibiotics

Rifaxamin is the most studied antibiotic. Three trials have been published [50–52]. One showed a decrease in transaminases [50], and one no change in ALT, HOMA, or steatosis, as assessed by MRS [51]. One study—a placebo-controlled study in patients with histologically confirmed NASH—found a positive effect on transaminase levels, disease activity as assessed by circulating levels of cleaved cytokeratin 18 (CK18), and insulin resistance [52]. These studies were small and limited, with no histological endpoints, and they do not permit firm conclusions.

### 3.1. Prebiotics, Probiotics, and Synbiotics

Prebiotics are nondigestible carbohydrates that stimulate the growth and activity of beneficial colonic bacteria. Prebiotics also produce changes in the composition or activity of the intestinal microbiome, which have a beneficial effect on host well-being and health.

Probiotics are microorganisms that can provide a positive effect on health when administered in an adequate dose. Probiotic therapy refers to the introduction of beneficial microorganisms into the intestinal flora. The most common strains of probiotics with beneficial health effects include *Enterococcus faecium*. *Bifidobacteria*, *Bacillus*, *Saccahaoromyces boulardii*, *Lactobacillus strains*, and *Lactobacillus*. The beneficial effects of probiotics include the production of butyrate and short-chain fatty acids and the production of stimulatory signaling proteins. In addition, there is a reduction in the secretion of pro-inflammatory cytokines and increases in mucin-2 expression, autophagy, and the upregulation of defensins [53].

Synbiotics are a combination of prebiotics and probiotics. The term refers to the synergism whereby a prebiotic component is selectively favored by the probiotic organism. This results in a more beneficial outcome. There is an improvement in the in vivo survival and activity of the probiotic and an enhancement of the benefit. Recently, the definition of synbiotics has been altered to include preparations favoring synergism as a result of probiotics metabolizing prebiotics to induce a rebalancing of dysbiosis in the gut and the host's health. The synergism of pre- and probiotics results in a selective stimulation of microbial growth and/or the activation of specific metabolism of the microbiome. The prebiotic component shields the probiotic from the effects of gastric acidity and proteolysis. It is important to select specific substrate and microbial combinations that can enhance the beneficial effects compared to probiotics or prebiotics alone.

The use of prebiotics, probiotics, and synbiotics have the potential to regulate the gut microbiota and have an influence on MAFLD. A systematic review of their effects on NAFLD and NASH has recently been published [54]. This review surveyed 13 trials with a total of 947 patients aged 18 to 80. Six studies involved probiotics, three involved prebiotics, and six involved synbiotics. A total of 11 trials were performed in patients with NAFLD and only 2 in patients with NASH.

There was a decrease in many inflammatory markers, including LPS, TNF-alpha, and IL-6, a decrease in liver enzymes, a decrease in serum lipids, a decrease in BMI, waist circumference, and HOMA-IR. In addition, there were decreases in the hepatic fatty liver index, as well as NAFLD fibrosis, and steatosis scores. There was also an increase in *Bifidobacterium levels*, and a decrease in *Clostridia* and *Erysipelotrichia* classes.

In summary, there is evidence to suggest a beneficial effect on MAFLD by the administration of pre-, pro-, and synbiotics, but further studies are required that include data on clinically important end-points. Recently, a systematic review of the effect of pre-, pro-, and synbiotics has been published [54,55].

### 3.2. Fecal Microbial Transplantation (FMT)

In view of the intimate association of the microbiome with metabolic syndrome, there have been studies assessing the effect of FMT on metabolic syndrome and/or MAFLD.

FMT was employed 1700 years ago during the Dong-jin dynasty in China for patients with severe diarrhea [56]. In the 16th century, during the Ming dynasty, Li Shizen also reported the use of yellow soup for the treatment of severe diarrhea [56]. In the 17th century, the Italian anatomist Fabricus Aquapendente employed fecal transplantation in veterinary medicine [57]. In the 1950s, at Johns Hopkins Hospital in Baltimore, fecal enemas were given to patients with pseudomembranous colitis [58]. More recently, Brandt et al., employed stool donated via a spouse and administered via a colonoscope for recurrent *Closteroides difficile* infection (CDI) [59], and FMT is now the accepted treatment for recurrent CDI [60].

The effect of FMT on MAFLD can be assessed by two main endpoints. The first is the effect of FMT on metabolic syndrome, which is the fundamental disturbance responsible for MAFLD. The second is the effect of FMT on the hepatic manifestation of MS-MAFLD and its clinically significant end-points.

### 3.3. Effect of FMT on Metabolic Syndrome and MAFLD

There are not many published studies on the use of FMT for treating metabolic syndrome or MAFLD. Of those published, there are discordances among the methods of administration and compositions of the fecal material administered, as well as the endpoints of the studies.

### 3.4. Effect of FMT on Metabolic Syndrome

FMT has been reported to influence body weight. In 2014, there was a case report of a woman with recurrent CDI who received an FMT from her 16-year-old daughter, who had a BMI of 26.4. The patient gained 34 pounds in weight and had a BMI of 33, 16 months after the FMT [61]. In 2012, a study by Vrieze et al. on 18 males with metabolic syndrome who received FMTs from healthy lean donors with a 6-week follow-up showed an increase in insulin sensitivity but no change in serum lipids or fasting glucose [44].

FMT from lean donors together with fiber supplementation has recently been shown to alter cardiometabolic outcomes. This study was a double-blind randomized controlled trial conducted on patients with severe obesity and metabolic syndrome. FMT was performed by the administration of oral capsules together with either high-fermentable or low-fermentable fiber supplements [62]. The main outcome studied was the effect on insulin sensitivity after 6 weeks based on the homeostatic model assessment (HOMA-2 IR/IS). Only the patients in the low-fermentable fiber and FMT group had improvements in HOMA2-IR at 6 weeks. No side effects were reported. The metabolic benefits were independent of diet or medications and were associated with altered microbial ecology and increased engraftment of donor microbes. Interestingly, the eradication of *Helicobacter pylori* in a study from China has been shown to decrease HOMA-IR, fasting blood glucose, triglycerides, HBA1c, and controlled attenuation parameter values (reflecting hepatic steatosis) [63].

The evidence for FMT on metabolic syndrome is inconsistent, however. FMT from a single lean vegan donor to 20 male patients with metabolic syndrome was shown to modulate the microbiome to a vegan profile, but no functional effects were found as assessed by trimethylamine-N-Oxide (TMAO) production or vascular inflammation parameters [64].

A meta-analysis of FMT in 42 randomized controlled trials of microbial therapy for the treatment of metabolic syndrome has recently been published [65]. The conclusion was that manipulation of the microbiome can improve fasting blood glucose, serum cholesterol and triglycerides, waist circumference BMI, HOMA, and diastolic blood pressure. There was, however, no effect on systolic blood pressure or HBA1c. Only four of the trials involved FMT.

The initial methods of the administration of feces involved invasive endoscopic procedures or nasogastric tubes. More recently, capsules containing lyophilized stools or fresh fecal material have become available [66]. This is more acceptable to patients and also enables the ability to easily repeat treatment rather than FMT being a "one and done" procedure.

A study of 22 patients with a BMI greater than 35 and without diabetes, NASH, or metabolic syndrome who received FMT in the form of capsules from a lean donor has been published [67]. This was well tolerated but did not reduce the BMI. The intestinal microbiomes and bile acid profiles became similar to those of lean donors.

Recently, the use of washed microbiota transplantation has been reported in China [68]. This is similar to traditional FMT, but the bacterial solution is prepared by an intelligent microorganism separation system. A recent report showed an improvement in metabolic syndrome using this method of fecal preparation. The protocol involves the application of washed fecal material via the upper GI tract (nasojejunal tube) or the lower GI tract (endoscopic intestinal tube). There are three courses of treatment. Each course consists of 120 mL of washed bacterial solution, once a day for three days. This is carried out once a month for three months, and then a further course is performed three months after the last dose. A study in South China on 237 patients with functional bowel disease, of whom 45 had MS and 195 did not, was carried out. WMT in the MS patients resulted in short-term (1 month after the first WMT), medium-term (2 months after the first WMT and long-term (6 months after the first WMT).

In the short term, there was a decrease in the fasting blood glucose, triglycerides, and BMI, together with an increase in HDL. In the medium term, there was a decrease in FPG, total cholesterol, LDL, non-HDL, and BMI. As a result of these changes, there was a decrease in the risk of patients for atherosclerotic cardiovascular disease [69]. This technique has also been found to reduce fasting plasma glucose in patients with high blood glucose [70].

Recently, a proposal to study the effect of FMT in patients with grade 1 hypertension has been initiated [71].

There may also be a role for targeting gut microbiota in the treatment of heart failure [72,73].

### 3.5. Effect of MFT on MAFLD

There are few clinical trials examining the influence of FMT on MAFLD that have been published. Animal studies have shown an effect of FMT on microbial dysbiosis in the gut [74–76]. The study by Zhou et al. [74] examined the impact of obesity on FMT for MAFLD. Colonoscopic administration of FMT from a healthy donor to patients with MAFLD followed by three consecutive daily fecal enemas found, at 1 month post-treatment, a decrease in liver fat and a reduction of gut microbacteria dysbiosis. In addition, the effect was more pronounced in patients with lean MAFLD compared to lean patients.

A randomized controlled trial has shown that FMT can reduce intestinal permeability (determined by the lactose/mannitol urine test) in patients with NAFLD [77]. This was a study of 21 patients with NAFLD who were allocated in a 3:1 ratio to receive allogenic (15) or autologous (6) FMT via endoscopy to the distal duodenum. A total of 7 of the 15 patients undergoing allogenic FMT had elevated small intestinal permeability that decreased on assessment 6 weeks after FMT. There were, however, no changes in the HOMA-IR or hepatic protein density fat fraction as assessed by MRI. Changes in intestinal permeability have a critical role in the development of MAFLD [78].

Another study of 61 obese patients with type 2 DM demonstrated a favorable hepatic outcome. The participants were divided into three groups: one receiving FMT from lean donors via gastroscopy and lifestyle changes, one with just FMT, and one with sham transplantation. The FMT was repeated every 4 weeks for 12 weeks. The flora of the recipients acquired the lean bacteria of the donors in 100% of the cases of FMT and lifestyle changes, 88.2% of just FMT, and 22% of the sham-transplanted participants. The FMT and

lifestyle change group achieved reduced total and low-density lipoprotein cholesterol and liver stiffness at week 24 compared to the baseline [79].

Currently, there are 11 trials registered on clinicaltrials.gov examining the effect of FMT on NAFLD (accessed on 27 April 2023). These are featured in the Table 1.

**Table 1.** Clinical Trials In Progress—FMT In NAFLD.

| | Title | Status | Interventions |
|---|---|---|---|
| 1 | Intestinal Microbiota Transplantation for Nonalcoholic Fatty Liver Disease | Unknown status | Other: intestinal microbiota transplantation |
| 2 | The Effect of Consecutive Fecal Microbiota Transplantation on Non-Alcoholic Fatty Liver Disease (NAFLD) | Unknown status | Other: Gut microbiome transplantation |
| 3 | Fecal Microbiota Transplantation (FMT) in Nonalcoholic Steatohepatitis (NASH). A Pilot Study | Unknown status | Drug: Fecal Microbiota Transplantation |
| 4 | Fecal Microbiota Transplantation for the Treatment of Non- Alcoholic Steatohepatitis | Not yet recruiting | Other: Fecal Microbiota Transplantation |
| 5 | Effects of Fecal Microbiota Transplantation on Weight in Obese Patients With Non-alcoholic Fatty Liver Disease | Unknown status | • Dietary Supplement: Diet<br>• Other: FMT<br>• Other: Physical Activity |
| 6 | Transplantation of Microbes for Treatment of Metabolic Syndrome & NAFLD | Completed | • Biological: Autologous<br>• Biological: Allogeneic |
| 7 | A Prospective, Randomized, Placebo-Controlled Pilot Study to Characterize the Intestinal Microbiome and to Evaluate the Safety and Fecal Microbiome Changes Following Administration of Lyophilized PRIM-DJ2727 or Placebo Given Orally in Subjects With Nonalcoholic Fatty Liver Disease | Not yet recruiting | • Drug: PRIM-DJ2727<br>• Drug: Placebo |
| 8 | Evaluate the Efficacy, Safety and Tolerability of Fecal Microbiota Transfer for the Treatment of Patients With Nonalcoholic Steatohepatitis | Not yet recruiting | • Drug: Group 1 or Experimental group<br>• Other: Group 2 or Control group |
| 9 | Synbiotics and Fecal Microbiota Transplantation to Treat Non- Alcoholic Steatohepatitis | Recruiting | Combination Product: LFMT-capsules |
| 10 | Dietary Counseling Coupled With FMT in the Treatment of Obesity and NAFLD—the DIFTOB Study | Active, not recruiting | Other: FMT and placebo |
| 11 | Fecal Microbiota Therapy Versus Standard Therapy in NASH Related Cirrhosis. | Unknown status | • Biological: Fecal Microbiota Transplant<br>• Other: Standard Medical Treatment |

Of these, five have unknown status, three are not yet recruiting, one is active, and one is completed. There are no published data on any of these trials.

FMT from healthy donors has been shown to influence hepatic gene expression. A study of 10 patients with NAFLD proven by biopsy who received FMT from vegan donors and had a repeat biopsy performed after 24 weeks showed a trend toward improved necro-inflammatory histology and significant changes in the expression of hepatic genes involved in inflammation and lipid metabolism compared to those participants who received an autologous transplant [80].

Recently, a randomized clinical trial of FMT in NAFLD was published [81]. This was a study of 75 patients with NAFLD. A total of 48 received FMT from donors who were "healthy undergraduate donors", but no mention was made of formal inclusion and exclusion criteria. The mode of administration of the allogenic stool was reported (in the abstract only) to be via colonoscopy and then three daily enemas. An increase in the

proportion of *Bacteroides* was seen in the patients receiving allogeneic FMT. No details were available for the clinical or laboratory parameters. This study recently received a critical commentary [82].

Another study that deserves to be mentioned, although not of FMT, is the effect of a freshwater fish-based diet compared to a freshwater fish-based and red meat-based diet for 84 days on liver steatosis and the microbiome. This study of 34 NAFLD patients showed a reduction in liver fat content as assessed by an MRI-proton density fat fraction (MRI-PDFF) of −4.89% in the freshwater fish diet versus −1.83% in the freshwater fish and red meat-based diet group [83].

A study from Israel on FMT in patients who have achieved rapid weight loss is intriguing [83]. Ninety patients with abdominal obesity or dyslipidemia who participated in a weight loss trial performed at their place of work were included in the trial. They were divided into three groups: Group 1 followed dietary guidelines, Group 2 followed the Mediterranean diet, and Group 3 followed a green Mediterranean diet. The green Mediterranean diet group received green tea and a supplement of *Wolffia globose* (Mankai strain). After 6 months, there was a mean weight loss of 8.3 kg. Fecal samples were obtained at this time and prepared as capsules. The study subjects were then divided into receiving capsules containing autologous feces (FMT–green Mediterranean) or placebo capsules until month 14 of the study. The FMT–green Mediterranean group had a smaller regain in weight compared to the placebo group (17.1% vs. 50% $p = 0.02$). There was also a smaller gain in the waist circumference and insulin rebound in the FMT–green Mediterranean group. In addition, there was a change in the IM bacterial population in this group.

### 3.6. Side Effects of FMT

FMT is an approved treatment for refractory recurrent *C. difficile* infection.

Although well tolerated, it is necessary to consider the side effects. A systematic review of FMT has been published by Wang et al. [84]. It covers 50 publications and a list of adverse effects judged to be likely to be related to the FMT in 42 of them. There were a total of 78 different adverse reactions, the commonest of which was abdominal discomfort. The incidence was higher when the feces was administered to the upper gastrointestinal tract compared to the upper tract (47.6% vs. 17.7%). Serious adverse effects were present in 2% of cases in the upper GI tract compared to 6.1% in the lower tract. The method of FMT may result in side effects, although the increasing use of capsules that are orally administered will likely result in a reduction of these episodes. There are also reports of autoimmune diseases following FMT. We have reported a case of immune thrombocytopenia (ITP) after both a first and second FMT from the same donor [85]. The importance of SCFA in the maintenance of normal intestinal role in humans is illustrated by our case report of starvation colitis in a hunger striker and its reversion by oral SCFA [86].

### 4. Conclusions

The intestinal microbiome has a very important role in the pathogenesis of metabolic syndrome and MAFL. Gut microbial dysbiosis is linked to the pathophysiology and progression of the disease. Manipulation of the microbiome by the administration of probiotics, prebiotics, or FMT has the potential to improve some of the parameters of MAFLD (Figure 2).

The most effective treatment for MAFLD is based on lifestyle changes, diet, and physical activity. The most urgent need is for the widespread adaptation of the necessary lifestyle changes that can decrease the prevalence of the disease, and in addition, its morbidity and rate of progression. In addition, it is possible to reverse the severity of hepatic fibrosis. Much work is necessary to fully understand the role of the individual bacteria, their interactions within the highly complex milieu of the intestinal microbiome, and the possible effects of fungi and viruses within the intestine. In addition, there is a role of genetics and downstream effects on inflammation, metabolites, and intestinal permeability.

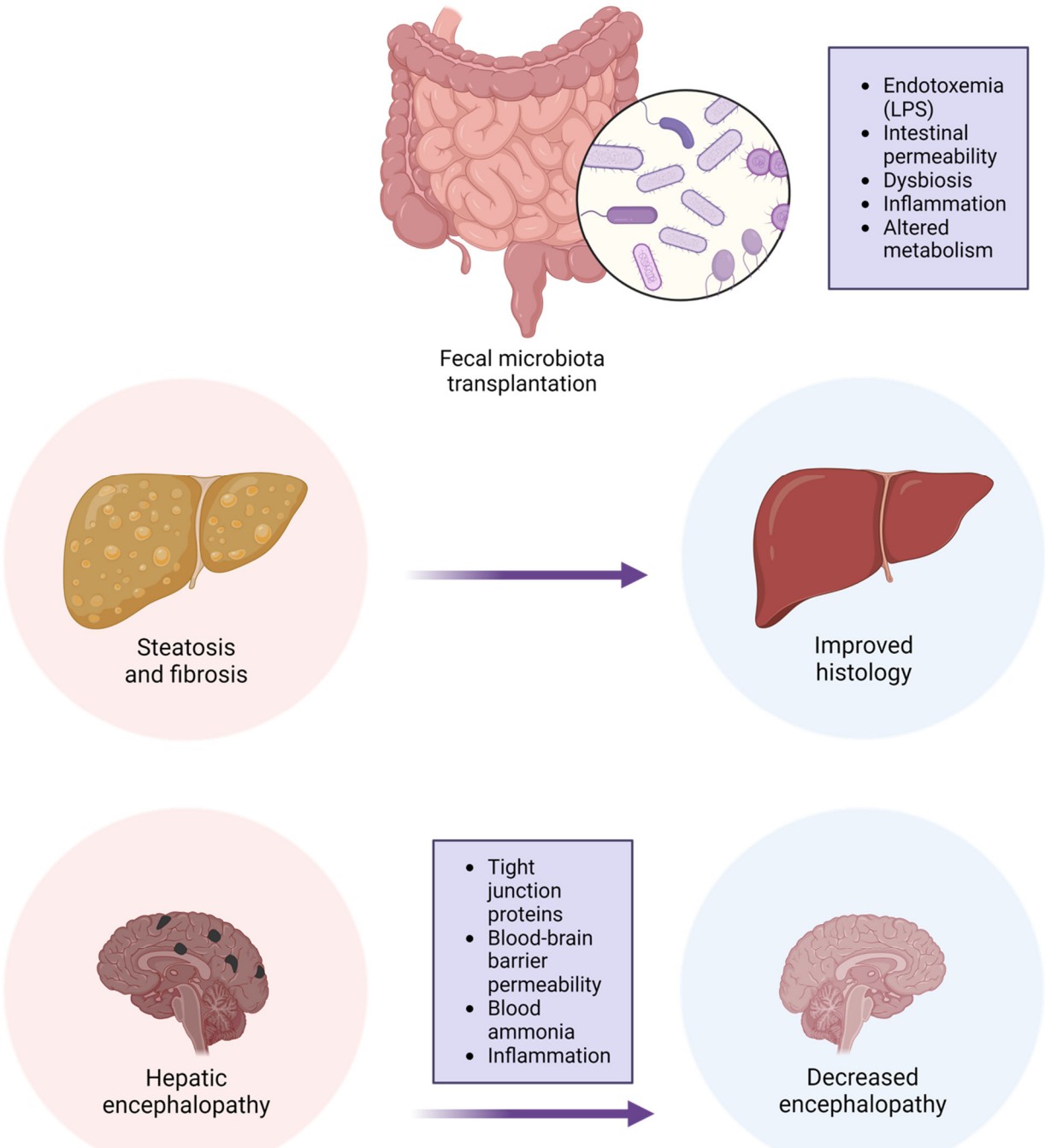

**Figure 2.** The potential effects of FMT on MAFLD.

Furthermore, it should be remembered that one of the main causes of the morbidity and mortality of MAFLD patients is cardiovascular events. It may be possible to individualize the composition of the microbiome to achieve a beneficial effect on both hepatic and cardiovascular outcomes.

It may be that the optimal treatment will require sophisticated modeling of the administration of probiotic strains and mixtures individualized for each patient. Recently, the FDA approved the use of VO, a fecal microbiota spore preparation for preventing the recurrence of *C. difficile* relapse after treatment.

FMT has some beneficial effects, but there is a need for robust controlled trials in order to determine the optimal method of administration, schedule, and course of treatment

of MAFLD. In addition, there may be potential to prevent MAFLD from developing by manipulation of the microbiomes of children and adolescents.

**Author Contributions:** S.D.H.M. and S.O.M. both initiated and planned and wrote the manuscript jointly. All authors have read and agreed to the published version of the manuscript.

**Funding:** This research received no external funding.

**Conflicts of Interest:** The authors declare no conflict of interest.

## Abbreviations

| | |
|---|---|
| ALT | Alanine aminotransferase |
| DM | Diabetes mellitus |
| FMT | Fecal microbial transplantation |
| GI | Gastrointestinal |
| GPR | glucagon protein-coupled receptors |
| HBA1C | Glycosylated hemoglobin |
| HOMA-IR | Homeostatic Model Assessment for Insulin Resistance |
| IM | Intestinal microbiome |
| ITP | Immune thrombocytopenia |
| MAFLD | Metabolic-associated fatty liver disease |
| MRS | Magnetic resonance spectroscopy |
| NAFLD | Non-alcoholic fatty liver disease |
| NASH | Non-alcoholic steatohepatitis |
| SCFA | Short-chain fatty acids |
| TMAO | Trimethylamine-N-oxide |
| WMT | Washed microbial transplantation |

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
