# Peer review of "The Intestinal Microbiome and the Metabolic Syndrome—How Its Manipulation May Affect Metabolic-Associated Fatty Liver Disease (MAFLD)"

_cimb, doi:10.3390/cimb45090455_

Round 1

Reviewer 1 Report

The manuscript is reasonably written, interesting and explores the influence of Intestinal Microbiome on Metabolic-Association Fatty Liver Disease and Metabolic Syndrome.

Some comments:

Keywords are missing.

There are several double spacing between words, throughout the text of the manuscript (namely, in the introduction, in lines 3, 6, 20, 23, etc.). Remove these double spacing throughout the document.

There are some formal issues that should be addressed.

Bibliographical references inserted in the text must have the following format “[1]”.

The table on page 8 must be numbered and the respective caption added.

The placement of figure 2 in the conclusions does not seem appropriate to me.

Congratulations on the study!

authors should consider doing an English quality review.

Author Response

We have changed the format of the references to (1).

We have changed the issues with the double spacing.

We have added in key words.

We have put a caption for Table 1.

We have made the reference to Figure 2 in the main section of the text.

Reviewer 2 Report

The review entitled "THE INTESTINAL MICROBIOME AND THE METABOLIC SYNDROME- HOW ITS MANIPULATION MAY AFFECT METABOLIC-ASSOCIATED FATTY LIVER DISEASE (MAFLD)" is very important because MAFLD or NAFLD/NASH pathogenesis and therapeutic strategy are still not clear enough. Disturbances in the intestinal microbiome have been shown to influence the development of MS, MAFLD, and their progression to fibrosis and cirrhosis are also still understudied. The authors concluded that  Gut microbial dysbiosis is linked to the pathophysiology and progression of the MAFLD. Manipulation of the microbiome by the administration of probiotics, prebiotics, or FMT has the potential to improve some of the parameters of MAFLD.

Major Comment:

In theIntroduction, the first paragraph should be  MS and MAFLD, the second IM, and the third Dysbiosis...

In MAFLD, the last paragraph (The metabolic syndrome is defined by obesity, .....) put before the paragraph (The pathophysiology of MS is not completely understood. ...). 

Describe in more detail the mechanisms of dysbiosis in the pathogenesis of MAFLD.

Prebiotics, probiotics, and synbiotics - should be prolonged (describe mechanisms of effects)

Put the meaning of abbreviations that are mentioned for the first time in the article, and continue to use abbreviations (MS, HOMA, ALT HDL........).

Check all manuscript for references (for example .. Bacterial overgrowth may develop as a consequence of intestinal tract secretions, including gastric and bile acid, mucin production, and gut antibacterial peptides. In addition, retrograde movement of bowel contents to the upper gut is inhibited by the ileocecal valve....)

Check all manuscript for grammar errors (space, new line...see paragraph Effect of MFT on MAFLD)

Author Response

We have made the changes in the order of the paragraphs.

We have expanded on the mechanisms of dysbiosis and also on pre pro and synbiotics.

We have added in more abbreviations.

Round 2

Reviewer 2 Report

After a thorough review of the revised manuscript, the authors have made suggested changes so that I consider this paper suitable for publication in CIMB